# Coinfection of Parvovirus B19 with Influenza A/H1N1 Causes Fulminant Myocarditis and Pneumonia. An Autopsy Case Report

**DOI:** 10.3390/pathogens10080958

**Published:** 2021-07-29

**Authors:** Domitille Callon, Fatma Berri, Anne-Laure Lebreil, Paul Fornès, Laurent Andreoletti

**Affiliations:** 1Cardiovir EA-4684, Faculty of Medicine, University of Reims Champagne Ardenne, 51097 Reims, France; fatma.berri@gmail.com (F.B.); anne-laure.lebreil@univ-reims.fr (A.-L.L.); pfornes@chu-reims.fr (P.F.); landreoletti@chu-reims.fr (L.A.); 2Pathology Department, Academic Hospital of Reims, Robert Debré, 51097 Reims, France; 3Virology Department, Academic Hospital of Reims, Robert Debré, 51097 Reims, France

**Keywords:** influenza A/H1N1, pneumonia, myocarditis, Parvovirus B19, case report

## Abstract

Parvovirus-B19 (PVB19) is a frequent causative agent of myocarditis. For unclear reasons, viral reactivation can cause acute myocarditis, a leading cause of sudden death in the young. Influenza A/H1N1(2009) virus (IAV/H1N1) is known for causing flu/pneumonia, but the heart is rarely involved. Co-infections of cardiotropic viruses are rarely reported and the mechanisms of viral interactions remain unknown. A 5-year old girl had a flu-like syndrome, when she suddenly presented with a respiratory distress and cardiac arrest. At autopsy, the lungs were found haemorrhagic. Lungs’ histology showed severe bronchiolitis, diffuse haemorrhagic necrosis, and mononuclear inflammation. In the heart, a moderate inflammation was found with no necrosis. IAV/H1N1 was detected in nasal and tracheal swabs, lungs, and the heart. The viral load was high in the lungs, but low in the heart. PVB19 was detected in the heart with a high viral load. Viral co-infection increases the risk of severe outcome but the mechanisms of interaction between viruses are poorly understood. In our case, viral loads suggested a reactivated PVB19-induced acute myocarditis during an IAV/H1N1 pneumonia. Viral interactions may involve an IAV/H1N1-induced cytokine storm, with a fulminant fatal outcome. Clinically, our case shows the importance of investigating inflammatory pathways as therapeutic targets.

## 1. Introduction

Human Parvovirus B19 (PVB19) is one of the most frequent causative agents of myocarditis, together with Enteroviruses and Herpes virus type 6 (HHV6) [1]. PVB19 is sometimes found in the heart as a bystander virus with no pathological effects [2]. The genome of PVB19 consists of a single-stranded DNA molecule of approximately 5.5 kb. Infection with PVB19 is common, with approximately 70–90% of adolescents having anti-PVB19 IgG detectable in serum. PVB19-infection is usually benign and in children it most commonly manifests with erythema infectiosum (fifth disease) [3]. Moreover, after an asymptomatic phase, for unclear reasons, reactivation of the virus can cause acute myocarditis, one of the leading causes of sudden death in the young [3,4,5]. Influenza A/H1N1 (2009) virus is known for causing flu and pneumonia, but the heart is very rarely involved [6,7]. Between 11 June 2009, and 10 August 2010, an influenza pandemic was declared by the World Health Organization. Worldwide, more than 13,500 patients died of influenza A⁄H1N1 (2009) infection. The death prevalence was similar in seasonal influenza as well as influenza A/H1N1 (2009) infections (0.3%), but the decedents’ characteristics were different. Influenza A/H1N1 (2009) involved mostly young apparently healthy patients [8]. Most of the deaths occurred in patients hospitalized with respiratory symptoms, fever, and digestive symptoms, including diarrhea and vomiting [9]. Detrimental inflammation of the lungs is a hallmark of severe influenza A virus infections [9]. Coinfections involving common cardiotropic and other viruses have rarely been reported. Viral co-infection increases the risk of severe outcome but the mechanisms of interaction between viruses are poorly understood [10]. Histology, immunohistochemistry, and virological molecular analysis are the gold standard in both endomyocardial biopsies and autopsy samples [10]. Frequent inadequate diagnostic tools explain underdiagnosis of myocarditis, especially in the young. When an autopsy is performed, lungs and heart tissues are rarely frozen, despite European guidelines [11]. We report on an autopsy case involving a 5-year old girl, who died of a co-infection involving a PVB19 myocarditis and an Influenza A/H1N1 pneumonia.

## 2. Case Presentation

A 5-year old girl was treated at home for a flu-like syndrome for 2 days with anti-inflammatory medications. Due to worsening breathing difficulties, her parents took her to the paediatric emergency room. The paediatricians noted pallor, asthenia, and thoraco-abdominal asyncrhony with chest indrawing. Pulmonary auscultation was abnormal. Neutrophils count was elevated with 6.56 G/L. She suddenly presented with a respiratory distress leading to a cardiac arrest. She had no diagnosis or medical history indicating diminished immunocompetence. 

At autopsy, the lungs were found haemorrhagic. There was no abscess. The heart and other organs were normal. Histological examination of lungs showed severe bilateral bronchiolitis, diffuse haemorrhagic necrosis, and mononuclear inflammation consisting of dense and diffuse macrophages and lymphocytic infiltrates (Figure 1A,B). In the heart, a moderate inflammation was found in both ventricles, characterized by T lymphocytes and macrophages infiltrates (Figure 1C,D). There was no necrosis. Other organs were normal.

Inflammatory blood markers (CRP: 37 mg/L and PCT: 4.71 μg/L) were increased at admission. Inflammatory cytokines (IFN-β, IL-6, MCP-1, TNF-α) were found increased in post-mortem lungs and heart homogenates in comparison with uninfected patient’s blood, indicating cytokine storm (Figure 1E). 

Nasal swabbing was performed for viral analysis according to published guidelines from U.S. Centers for Disease Control and Prevention (CDC protocol of real-time RT-PCR for influenza A H1N1 (2009)). Cerebrospinal fluid and blood were collected for bacterial analysis. Fresh and frozen lung samples were collected for bacterial and viral analysis, respectively. Frozen heart samples were collected for viral analysis according to European guidelines [10]. No bacterial agent was found by PCR (M-DiaMult™, Diagenode^®^ for Neissera meningitidis, Chla/Myco pneumo r-gene^®^ for Chlamydia pneumonia and Mycoplasma pneumoniae, Bio-Evolution^®^ for Bordetella pertussis) in nasal swabs and bronchial aspiration. Cultures for aerobic or anaerobic bacteria were negatives in blood and cerebrospinal fluid. 

After tissue sample lysis, the NucliSENS easyMAG^®^ instrument (BioMérieux, Lyon, France) was used for total nucleic acid extraction, according to the manufacturer’s protocol. Two RT-PCR DNA microarray detection systems were used in combination for the detection of human respiratory viruses in the frozen heart and lung samples. Clart Pneumo Vir and Clart FluA Vir kits (Genomica, Madrid, Spain) allowing simultaneous detection of 21 different types and subtypes of human respiratory viruses (influenza A virus [seasonal A/H1N1 and A/H3N2, and new influenza A/H1N1 (2009) virus strains], influenza B virus, influenza C virus, parainfluenza virus 1, 2, 3, 4, 4A and 4B, respiratory syncytial virus A and B, rhinovirus, adenovirus, enterovirus type B, bocavirus, coronavirus E-229, metapneumovirus A and B) were used according to the manufacturer’s protocol. Influenza A/H1N1 (2009) virus was detected in all samples (nasal swabs, tracheal swabs, lungs, and heart). 

Real-time RT-PCR assays (SuperScript III Platinum one-step quantitative RT-PCR system; Invitrogen, Carlsbad, CA) were performed for the identification and quantification of the influenza A/H1N1 virus. The virus was identified in all samples. The viral load was 611 genome copies (gc)/μg of nucleic acids extracted in the lungs, 6 and 20 gc/μg nucleic acids extracted, respectively, in the left ventricle (LV) and the right ventricle (RV) (Figure 2A). 

Real-time PCR and RT-PCR assays for herpesviruses (HSV-1, HSV-2, VZV, CMV, EBV, and HHV6) and other cardiotropic viruses (PVB19, Enterovirus) were used on frozen heart and lung samples (Clart^®^ Entherpex kit, Genomica, Madrid, Spain; PVB19/HHV6/Enterovirus r-gene^®^, Argène bioMerieux, Verniolle, France). PVB19 was detected in the heart with 4.4 × 10^6^ gc/µg of extracted DNA in the LV and 3.83 × 10^6^ gc/µg of extracted DNA in the RV, but not in the lungs. HHV6 was detected in the heart but with a non-significant viral load (Ct > 35 or undetermined). 

## 3. Discussion

Thorough viral molecular analysis showed a co-infection PVB19—Influenza A/H1N1 virus as the cause of death of the 5-year old girl. She had no past medical history indicating diminished immunocompetence. The fulminant fatal outcome following a 3-day flu-like syndrome raises the question of an interaction between PVB19, a frequent cardiotropic virus, and influenza A/H1N1 virus, a pulmonary agent.

Influenza A genomic RNA load was significantly higher (>2log10) in the lungs than in the heart and was responsible for pulmonary damages (Figure 2A). Diffuse alveolar damage is the most common pathologic finding in Influenza A/H1N1 infection [7]. Acute necrotizing trachea-bronchitis or bronchiolitis is also a frequent finding [8,9,12,13,14]. In our case, the microscopic lung findings were in accordance with previously reported Influenza A/H1N1 infections (Figure 1A,B). Influenza A/H1N1 (2009) virus was the only cause of pulmonary damages in our case, as demonstrated by thorough microbiological investigations on fresh and frozen lung samples. Recent recommendations by the ESCMID Study Group for Forensic and Postmortem Microbiology (ESGFOR) advocate for thorough quantitative virological analysis of lung and heart frozen samples, in autopsy cases [11].

Influenza A/H1N1 (2009) has also been reported as a rare cause of myocarditis [7,9,15,16,17]. Influenza A/H1N1 (2009) was detected by PCR in blood, nasal swabs or formalin fixed paraffin embedded samples [7,15,16,17]. However, in most reports, the possible presence of other cardiotropic viruses was not investigated.

PVB19 DNA was detected only in the heart (Figure 2B). Interpretation of viral markers remains a difficult task. Viruses can be an incidental finding in autopsy studies of individuals who died of non-cardiac death. In particular, PVB19 and HHV6 have been reported to be frequent bystanders with no clinical consequences [2,3,18]. Distinction between active viral infection in target organs and a bystander virus requires quantification of genomic viral load in frozen samples [3,4]. HHV6 was a bystander virus in our case, since viral load was very low (CT > 35 in left ventricle and undetermined in right ventricle) [19]. We used a published cutoff point of 500 genome copy per microgram of nucleic acids to distinguish between active and latent PVB19 infection [3]. The very high viral load indicates a crucial role of PVB19 in the cause of death, in addition with Influenza A/H1N1-induced pneumonia. High PVB19 load has been shown to be linked to sustained inflammation in myocarditis and dilated cardiomyopathy [20]. However, comparison with PVB19 loads in normal heart and non-cardiac deaths is lacking in most studies [20]. 

Our case raises the issue of the interaction of PVB19 and Influenza A/H1N1 viruses in causing pneumonia and myocarditis. PVB19 is a frequent cause of myocarditis, while Influenza A cardiac infection is uncommon [3,15,21]. The link between influenza virus viral load in respiratory samples and disease severity is not clearly established. High Influenza A/H1N1 viral load was found correlated with abnormal findings on chest X-ray and pneumonia in children [22,23,24]. Moreover, Influenza A/H5N1 virus high viral load was found correlated with high mortality [25]. Large cohort studies are mandatory to better understand the link between Influenza virus viral load and disease severity. Comparisons of Influenza virus viral loads between multiple organs samples, such as in our case, could help in understanding replication fitness and pathogenesis in various cell types.

The low Influenza A/H1N1 genome copy number in cardiac tissues associated with a significant PVB19 genome copy number supports the hypothesis of a reactivated PVB19-induced acute myocarditis during an Influenza A/H1N1 infection, as reported for other viruses associated with PVB19: CMV [26], HHV6 [27], EBV [28], HIV [29], and IAB [30]. However, PVB19 coinfection with Influenza A virus has not been reported. Immunosuppression or modulation are known to be the main factors of PVB19 reactivation [28,31]. In our case, the girl had no diagnosis or medical history indicating diminished immunocompetence. The fulminant fatal outcome suggests a cytokine storm induced by Influenza A/H1N1 infection, which could have triggered the reactivation of a PVB19 latent cardiac infection, as reported in other viral infections [32]. PCT level increase has been reported in severe viral infection with no bacterial agent, such as in our case [33]. However, there is no way to rule out the possibility that the patient became newly infected with PVB19 during an Influenza A infection. Consistently, inflammatory cytokines were increased in the lungs and the heart homogenates, in comparison with uninfected patient’s blood (Figure 1E). 

In addition to further clinical cases with thorough molecular virological analysis, in vitro studies will be helpful in better understanding the mechanism of viral interactions and the role of viral-induced inflammation, a clinical therapeutic target.

## Figures and Tables

**Figure 1 pathogens-10-00958-f001:**
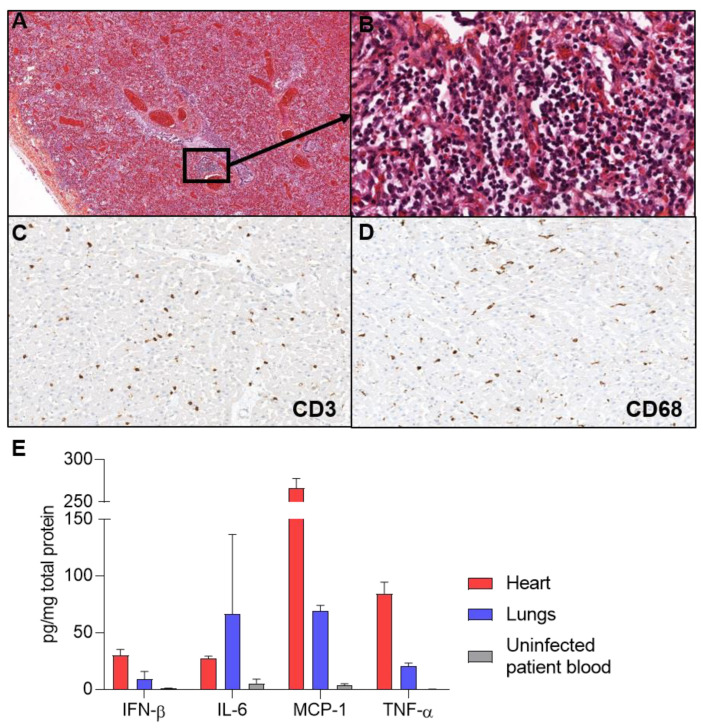
Pneumonia and myocarditis. (**A**) Hematoxylin, eosin, and safran-stained sections show scattered foci of necrosis and inflammation (×40) (**B**) At higher magnification of an inflammatory Figure 400. (**C**) CD3 immunostaining revealed infiltrates of T lymphocytes in the heart (×400). (**D**) CD68 immunostaining revealed macrophages cells infiltrates in the heart (×400). (**E**) Inflammatory cytokines were quantified by ELISA assays, revealing IFN-β, IL-6, MCP-1, and TNF-α increase compared to uninfected patient blood.

**Figure 2 pathogens-10-00958-f002:**
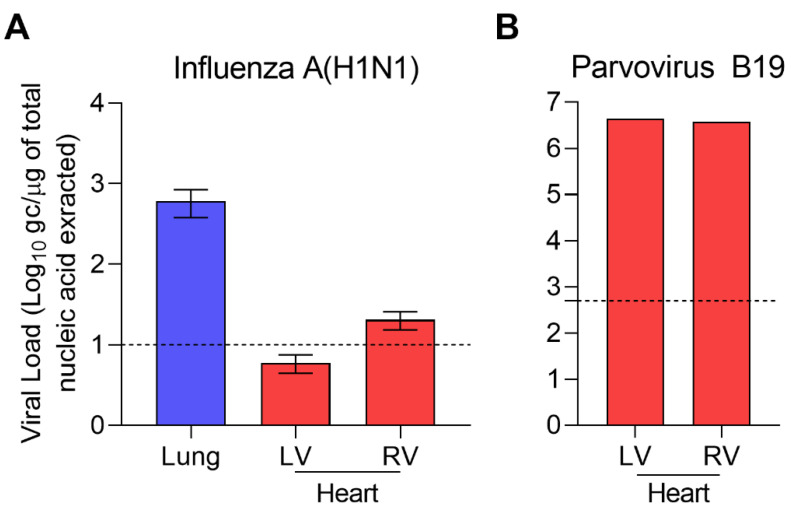
Molecular biology revealed Influenza A(H1N1) and Parvovirus B19 co-infection. (**A**) Influenza load was 611 genome copies (gc)/μg of nucleic acids extracted in the lungs, 6 and 20 gc/μg nucleic acids extracted respectively in the left ventricle (LV) and the right ventricle (RV). (**B**) PVB19 was detected in the heart with 4.4 × 10^6^ gc/µg of extracted DNA in the LV and 3.83 × 10^6^ gc/µg of extracted DNA in the RV. Dotted lines represented threshold for significant replicative activity.

## Data Availability

The data presented in this study are available on request from the corresponding author. The data are not publicly available due to ethical issues (patient identification).

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
