# Peer review of "Coinfection of Parvovirus B19 with Influenza A/H1N1 Causes Fulminant Myocarditis and Pneumonia. An Autopsy Case Report"

_pathogens, 2021, doi:10.3390/pathogens10080958_

Round 1
Reviewer 1 Report
The manuscript under review by Callon et al details a case report of a fatal dual infection of influenza virus (IAV) and parvovirus B19 (PVB19) in a pediatric patient. Often cases of multi-viral infections such as the case detailed by Carron et al are overlooked or under investigated. The authors should be commended for their due diligence and for drafting a case report for publication. The report has scientific merit and should be considered for publication. Below are several comments meant to improve the manuscript for eventual publication.
- The authors should consider revising the title. In particular, remove the word “interaction”. The word “interaction” is too general a term and has no specific meaning. It only acts to make the title ambiguous. The new title should be more descriptive of the case being reported.
- The introduction is limited and not very informative. It maybe in it’s current state due to word or character limits but the authors should consider revising the introduction.
- The figure legends read more as results or discussion content and repeat what is already in the text. The figure legends should just provide descriptions of the data presented in the figure.
- The hypothesis that the PVB19 infection is an example of reactivation due to IAV infection (Lines 148-156) is valid but there is no way to rule out the possibility that the patient became newly infected with PVB19 during a IAV infection. The authors should address this in the discussion as an alternative explanation of the case.
- Several times in the manuscript (Line 49, 107, and 151), the authors describe the autopsy subject as “immunocompetent”. The patient may have no diagnosis or medical history indicating diminished immunocompetence but that is not the same thing as an affirmative confirmation that the patient was in fact immunocompetent. She may have suffered from an undiagnosed or unknown condition. The authors should replace the statements that “She was immunocompetent” with more accurate descriptors such as “she had no diagnosis or medical history indicating diminished immunocompetence”
Below are several optional suggestions that could improve the manuscript.
- For Fig 1 A and B, the inclusion of low power image of the lung tissue with higher powered insets would improve the figure. It would give the reader a better sense of the lung pathology.
- Since the authors have access to FFPE lung and heart tissue, immunofluorescent staining for IAV proteins would be very valuable data and could be used to further support the assertion of active IAV infection in the heart.
Reviewer 2 Report
In this case report, Callon et al have reported the co-infections of Influenza virus and Parvovirus B19 in a 5 year old patient with severe outcome of myocarditis and pneumonia.
Major comments:
- Title is misleading. Author has not studied the mechanism of interaction between influenza and parvovirus B19, while co-infection of both viruses was observed in this particular case. It is strongly recommended to change the title representing the co-infections of both viruses that lead to severe outcomes such as pneumonia and acute myocarditis in immunocompromised patient.
- Page 2, line 42. Remove the statement “Interaction between the viruses is examined.” As mentioned above, the author has determined viral aetiology in two different tissues but did not study a particular mechanism that shows the association between them.
- Reference 10 given for the statement “Frozen heart samples were collected for viral analyses according to European guide” is not appropriate. Provide the right reference.
- References (4, 26, 27) given in support of the statement (The low Influenza A/H1N1 genome copy number in cardiac tissues associated with a significant PVB19 genome copy number support the hypothesis of a reactivated PVB19-induced acute myocarditis during an Influenza A/H1N1 infection) are not correct. Please remove this statement or provide the right references that show the association of influenza infection with the reactivation of PVB19.
- It’s very strange that the PCT level was detected quite high (4.71 μg/L), still no bacterial infection seen in patient’s samples. Explain.
- What bacterial agents were tested in blood and cerebrospinal fluid (page 3, line 73)? Please include in the text.
Minor Comments:
- In figure 1, the author should rewrite legend for each figure separately by clearly mentioning how figure A differs from figure B and likewise what Figure C and D represents. Also mention what CD3 and CD68 represents here that are also not mentioned in the text.
- However, parvovirus B19 is a cardiotropic virus but I am wondering if the author has detected Parvovirus B19 in lung as well?
- Word “analyses” should be replaced with “analysis, which is represented throughout text as Viral analyses/bacterial analyses/molecular analyses.
- What “FFPE” represents (line 122) in text?
Round 2
Reviewer 2 Report
Below are a few points that should be addressed before considering the article for final submission .
- Author has mentioned that they have added below reference in support of increased PCT level without detecting bacterial infection. I think the author forgot to include this in the text as well as in the reference list. Please update the text and reference list.
“Elevated PCT level has been reported in severe viral infection: Gautam S, Cohen AJ, Stahl Y, et al. Severe respiratory viral infection induces procalcitonin in the absence of bacterial pneumonia. Thorax 2020;75:974-981. We added the reference (Page 6, line 174).”
- Author has included the name of bacterial agents but forgot to mention what kit/PCR primers were used for PCR detection of these agents (page 3, line 86). Please add this information in the text.
- I am wondering if the author has performed PVB19-specific serological diagnosis (IgG and IgM ELISA). In this way, it can be confirmed if a patient was recently infected (IgM positive) or the patient became positive due to actual reactivation of PVB19 (IgM negative but IgG positive). Please include serology data in paper or justify this.
- In below article, co-infection of influenza B virus with parvovirus B19 has been earlier reported in 11 year old patient with secondary bacterial infection. It is suggested to include this reference in the text.
https://pubmed.ncbi.nlm.nih.gov/14556063/ (Krell S, Adams I, Arnold U, Kalinski T, Aumann V, König W, König B. Influenza B pneumonia with Staphylococcus aureus superinfection associated with parvovirus B19 and concomitant agranulocytosis. Infection. 2003 Oct 1;31(5):353-8.)
Author Response
Please see the attachment.

This manuscript is a resubmission of an earlier submission. The following is a list of the peer review reports and author responses from that submission.